# Prognostic Value of B7-H3 and a Novel Scoring System in Localized Renal Cell Carcinoma

**DOI:** 10.3390/medicina61050867

**Published:** 2025-05-09

**Authors:** Faruk Recep Özalp, Kutsal Yörükoğlu, Eda Çalışkan Yıldırım, Mehmet Uzun, Erkut Demirciler, Hüseyin Salih Semiz

**Affiliations:** 1Department of Medical Oncology, Faculty of Medicine, Dokuz Eylul University, 35340 Izmir, Turkey; caliskan_eda@yahoo.com (E.Ç.Y.); erkutdemirciler@gmail.com (E.D.); hsalihsemiz@hotmail.com (H.S.S.); 2Department of Pathology, Faculty of Medicine, Dokuz Eylul University, 35340 Izmir, Turkey; kutsal.yorukoglu@deu.edu.tr

**Keywords:** RCC, prognostic scores, B7-H3, UISS, GRANT

## Abstract

*Background and Objectives*: Renal cell carcinoma (RCC) is a biologically heterogeneous malignancy, and traditional prognostic models often fail to provide accurate risk stratification. B7-H3 (CD276), an immune checkpoint molecule, has been implicated in RCC progression but remains underexplored as a prognostic biomarker. *Materials and Methods*: This retrospective study analyzed 52 patients with localized RCC who underwent nephrectomy. Immunohistochemical staining was used to assess B7-H3 expression. A novel prognostic scoring system, the Renal Immune Prognostic Index (RIPI), incorporating B7-H3 expression, tumor necrosis, tumor grade, and pathological staging, was developed and validated. Kaplan–Meier survival analysis and Cox proportional hazard models were employed to evaluate disease-free survival (DFS) and overall survival (OS). *Results*: High B7-H3 expression was significantly associated with shorter DFS (12 vs. 54 months; *p* = 0.001) and OS (70 vs. 123 months; *p* = 0.002). The RIPI demonstrated strong prognostic performance, stratifying the patients into distinct risk groups with a C-index of 0.82. The high-risk patients had a median DFS of 14 months, compared with 125 months in the low-risk group (*p* < 0.001). *Conclusions*: B7-H3 expression serves as a significant prognostic biomarker in localized RCC, correlating with poorer survival outcomes. The integration of B7-H3 into the RIPI enhances risk stratification by incorporating both molecular and pathological features. These findings support the incorporation of immune biomarkers into clinical practice and highlight B7-H3 as a potential target for novel therapeutic strategies in RCC.

## 1. Introduction

Renal cell carcinoma (RCC) is one of the most lethal urological malignancies, accounting for approximately 2–3% of all cancers globally, with higher incidence rates observed in men and the elderly [1,2]. RCC encompasses a variety of histological subtypes, with clear-cell renal cell carcinoma (ccRCC) being the most common. Recurrence or disease progression occurs in 30–40% of patients following surgery, resulting in poor long-term survival [3,4]. Advanced RCC further complicates management and typically does not extend progression-free survival (PFS) beyond 24 months, even when using a combination of molecularly targeted therapies and immune checkpoint inhibitors [5,6].

In localized RCC, systems such as the tumor-node-metastasis (TNM) staging system and the Fuhrman grading system were mostly used to determine the at-risk population. However, these systems often fail to predict recurrence-free survival (RFS) or OS accurately because of RCC’s biological heterogeneity. To address this limitation, several integrated prognostic models, including the Mayo Clinic SSIGN score, the UCLA Integrated Staging System (UISS), and the GRANT score, have been developed. These models aim to improve predictive accuracy by combining various clinicopathological factors [7,8,9]. The UISS score combines the pathological stage, Eastern Cooperative Oncology Group (ECOG) performance status, and tumor grade to predict OS and PFS. The GRANT score combines similar parameters but places more emphasis on histological features, aiming to improve risk stratification for recurrence and metastasis. However, these models still lack incorporation of molecular biomarkers that could refine patient risk stratification further [10,11].

Given the rising costs and limited availability of healthcare resources, there is a critical need to identify patients who will benefit most from intensive surveillance and therapeutic interventions, especially in the context of emerging adjuvant therapies. Biomarker research therefore plays a crucial role in optimizing resource allocation and tailoring treatment strategies more effectively [12,13,14]. Combining molecular markers and immune-related markers with established prognostic models such as SSIGN, the UISS, and GRANT may improve patient stratification, but further validation is required for optimal clinical application [15].

B7-H3 is a member of the B7 family of immune checkpoint molecules, which play significant roles in regulating immune responses [16]. It is expressed on various cancer cells and has been implicated in immune evasion, tumor progression, and metastasis [17]. In RCC, B7-H3 overexpression has been linked to a poor prognosis, making it a potential biomarker for risk stratification and a therapeutic target. Its ability to modulate immune responses within the tumor microenvironment is primarily responsible for these associations [18,19].

In this study, we investigate B7-H3 as a potential prognostic indicator for localized RCC. The aim of this study is to improve risk stratification by integrating B7-H3 into clinicopathological data and to create a scoring system.

## 2. Materials and Methods

### 2.1. Patient Characteristics and Study Design

This retrospective cohort study was approved by the Dokuz Eylül University Ethics Committee (9021-GOA), with all participants or their guardians providing written informed consent. Between 2016 and 2023, RCC patients followed in the Medical Oncology Clinic of Dokuz Eylül University Hospital were screened. Out of 184 patients identified, 118 were excluded due to de novo metastatic disease, and 14 were excluded because their pathology specimens were unsuitable for immunohistochemical staining. Consequently, the study included 52 patients with various RCC subtypes, including clear-cell (*n* = 33), papillary (*n* = 13), and other (*n* = 6) (Figure 1). The clinicopathological and treatment-related characteristics of these patients, such as age at diagnosis, performance status, tumor histology, and GRANT and UISS scores, were retrospectively reviewed using patient files and electronic medical records.

### 2.2. Immunohistochemical Analysis

Tissue microarrays (TMAs) were constructed using 3 mm tumor punches from radical or partial nephrectomy specimens. Immunohistochemical staining was conducted on the Ventana BenchMark XT platform with the B7-H3 monoclonal antibody (ANTI-CD276 EA, MA5-15693, Thermo Fisher Scientific, Waltham, MA, USA). The analysis focused on the membranous staining of tumor cells. B7-H3 expression was assessed by an experienced pathologist (K.Y.) blinded to patient treatment, survival, and demographic data. Scoring was based on staining intensity and completeness: a score of 3+ was assigned for complete and intense membrane staining in at least 10% of tumor cells, 2+ for weak to moderate intensity or incomplete membrane staining, 1+ for weak intensity, and 0 for the absence of staining; following the method described by Inamura et al. to ensure scoring reliability [20], a subset of 10 patients were scored twice, demonstrating a high kappa value (>0.61), indicating substantial agreement (Figure 2). Following this validation, general scoring was completed for all samples. The 0/1+ group was categorized as low expression group, and 2+/3+ group as high expression group.

### 2.3. Statistical Analysis

Descriptive statistics were utilized to analyze the clinicopathological and treatment-related features of the patients. Categorical variables were presented as percentages, while continuous variables were presented as medians and ranges. The duration from surgical treatment to the first evidence of recurrence (or death) was defined as DFS, while OS was defined as the duration from diagnosis to death from any cause. Survival was assessed using the Kaplan–Meier method and compared between the groups using the log-rank test. The median follow-up period was determined using the reverse Kaplan–Meier method. Unadjusted hazard ratios (HRs) for DFS and OS were calculated using Cox proportional hazard regression models. To address potential confounding factors, adjusted HRs were determined using multivariate regression analysis.

A new DFS-based risk scoring system (RIPI) was developed based on the following five factors: B7-H3 expression, tumor necrosis and pT1/2-3/4 group, pN group, and tumor grade group. The formula was constructed using hazard ratios as multipliers and is defined in Figure 3. Harrell’s concordance index analysis was used to compare UISS, GRANT, and RIPI. While RIPI was divided into 2 groups as low- and high-risk, Youden index was utilized in the receiver operating characteristic (ROC) curve. The statistical analyses were conducted using the SPSS Statistics 25.0 for iOS software program (SPSS, Inc., Chicago, IL, USA), and a significance level of *p* < 0.05 was applied. Harrell’s C-index was calculated with R software version 2024.12.1+563 (The R Foundation for Statistical Computing, Vienna, Austria).

## 3. Results

The purpose of this study was to evaluate the clinicopathological characteristics of 52 patients with localized RCC based on B7-H3 expression levels. Clinicopathological and treatment-related characteristics are summarized in Table 1. The cohort of patients included 33% (*n* = 17) women and 67% (*n* = 35) men with a median age of 65 ± 11 years. In 31% (*n* = 16) of the cases, the B7-H3 expression was 0, while the expression levels in the remaining samples were as follows: 38.5% (*n* = 20) of the participants scored +1, 25% (*n* = 13) +2, and 5.7% (*n* = 3) +3.

The median follow-up was 64 months (CI 33.8–94.1). The median DFS for the entire cohort was 29 months (CI 16.03–41.9). The patients with low B7-H3 expression had a significantly longer DFS compared to those with high expression (54 months vs. 12 months, *p* = 0.001). Univariate analysis showed that tumor necrosis, regional lymph node involvement, higher tumor grade, advanced pT stage, presence of lymphovascular invasion, and high B7-H3 expression were associated with shorter DFS. The median DFS for patients with tumor necrosis was 8 months, which was significantly shorter than that for patients without necrosis (54.0 months; *p* < 0.001). Similarly, grade 3/4 tumors were associated with significantly shorter DFS (12 months) compared to grade 1/2 tumors (57 months, *p* < 0.001). Furthermore, DFS was shorter in the pT3/4 group than in the pT1/2 group (17 vs. 62 months, *p* < 0.001). In the univariate and multivariate analyses, age, gender, Karnofsky PS, pathology group, and sarcomatoid differentiation were not found to significantly affect DFS (Table 2).

The median OS for the entire cohort was 112 months. Low B7-H3 expression was associated with a significantly better prognosis, with a median OS of 123 months compared to 70 months in the high-expression group (*p* = 0.002). In addition, tumor necrosis and grade group had a significant negative impact on OS. The median OS for patients with necrosis was 31.6 months, while the median OS for patients without necrosis was not reached (*p* < 0.001). In the high-grade group, the median OS was 69 months, while the median OS was not reached in the low-grade group (*p* = 0.003).

The C-index of the RIPI was found to be 0.82 (CI 0.75–0.88). The C-index values of the UISS and GRANT scores were 0.71 (CI 0.63–0.78) and 0.69 (CI 0.60–0.78), respectively. The RIPI was divided into two groups as high (*n* = 29) and low (*n* = 23), with a cut-off value of 2.79 according to the ROC curve analysis. The median DFS was 125 (CI 15.4–234.5) months in the low-risk RIPI group and 14 (CI 4.7–23.2) months in the high-risk group (*p* < 0.001) (Figure 4).

These results highlight the significant prognostic value of B7-H3 expression, tumor necrosis, pT group, pN group, and tumor grade in localized RCC and demonstrate the effectiveness of the proposed scoring system in classifying patients according to their likelihood of recurrence.

## 4. Discussion

In this study, which included patients with RCC, disease recurrence occurred in 73.1% of the patients during a median follow-up period of 64 months. A significantly shorter DFS was observed in the high B7-H3 group, and the B7-H3 group was able to independently predict the DFS with an HR of 2.78 in the poor group. By integrating B7-H3 expression and pathological features (tumor necrosis, pathological T stage, and tumor grade), we developed the RIPI, a novel risk stratification tool that successfully differentiates patients into distinct prognostic categories.

As an immune checkpoint protein, B7-H3 binds to receptors on T-cells and other immune cells, inhibiting their activity and allowing the tumor to escape immune surveillance [21,22]. This immune evasion contributes to the aggressive nature of RCC and the poor survival outcomes observed in patients with high B7-H3 expression. Given the growing evidence for the importance of B7-H3, incorporating its expression into clinical decision making may lead to more personalized treatment strategies.

A number of studies have been published that examine the relationship between B7-H3 expression and survival. Yu et al. demonstrated that an increase in B7-H3 expression was associated with a reduced time to relapse [23]. A similar association was observed by Iida et al., who reported that increased B7-H3 expression was associated with shorter DFS in RCC [14]. Mischinger et al. demonstrated a negative correlation between B7-H3 expression and OS [24]. Our survival findings reinforce these observations and further highlight the prognostic utility of B7-H3 in localized RCC. Other factors affecting DFS other than B7-H3 were tumor grade group, pT, pN, and the presence of necrosis. Performance status and sarcomatoid differentiation were not found to significantly affect DFS. The effect of performance status on DFS may be more pronounced in metastatic disease as it helps to determine the treatment modality. Patients undergoing nephrectomy for local disease are expected to have a certain ECOG performance status. This may lead to patient selection bias from the onset. The fact that sarcomatoid differentiation was not associated with DFS may be related to the small number of patients in the study population and the low rate of sarcomatoid differentiation.

Identifying the high-risk group is very important in RCC. Close follow-up of the high-risk group can help us to detect recurrence early and recruit more patients for metastasectomy. In addition, adjuvant pembrolizumab treatment has been shown to reduce the risk of recurrence in the high-risk population, as determined by UISS and Leibovich score. Although high-risk groups have traditionally been defined using clinicopathological scoring, new markers are needed to identify the potential immunotherapy-resistant subgroup.

The European Association of Urology recommends the UISS and GRANT scores as prognostic scoring systems independent of the pathological subtype in patients with clear-cell, papillary, and chromophobe RCC. The Leibovich and VENUSS scores are valid for specific pathological subtypes. The GRANT and UISS scores have been analyzed in many patient populations. In their study, Cortellini et al. found no significant correlation between DFS and UISS score, whereas they found a significant effect of GRANT score on DFS. In addition, the C-index of the UISS score was found to be lower than that of the GRANT score (0.51 vs. 0.59) [25]. In our study, all the prognostic scores were found to be significant for DFS. Comparing GRANT and UISS scores in patients with disease progression, Giancristofaro et al. found a similar C-index of 0.75 vs. 0.76 [26], and Rosiello et al. found an AUC of 0.718 vs. 0.839 in clear-cell RCC and 0.846 vs. 0.795 in papillary RCC [27]. When we compared the C-indexes, the C-index of the RIPI was higher than the UISS and GRANT. This may be because our patient population was heterogeneous and represented a small number of patients. The C-index of the UISS score was higher than that of GRANT. Although ECOG PS was not significant in multivariate analysis, this result may be due to the fact that the patient population in the study was predominantly clear-cell RCC. The RIPI may be an important scoring system in the adjuvant pembrolizumab era as it contains a parameter such as B7-H3, which may be associated with resistance to immunotherapy in the metastatic stage.

B7-H3 is an accessible and cost-effective biomarker that can be easily evaluated using immunohistochemistry. The ability to incorporate B7-H3 staining into existing IHC panels without significant additional cost or effort makes it a highly practical candidate for routine clinical use. Identifying patients with high B7-H3 expression may be crucial for stratifying those who are less likely to benefit from immune checkpoint inhibitors. If B7-H3 expression is indeed inversely correlated with IO response, it could serve as a predictive biomarker, guiding treatment decisions. Targeting B7-H3 with monoclonal antibodies could potentially enhance anti-tumor immune responses, reversing the immune suppression within the tumor microenvironment [28]. Early-phase clinical trials targeting B7-H3 are underway, exploring its potential in combination with PD-1, which have revolutionized cancer immunotherapy [29]. Patients with high B7-H3 expression might be better candidates for combination strategies, such as pembrolizumab plus tyrosine kinase inhibitors (TKIs) or even pembrolizumab plus anti-B7-H3 therapies in adjuvant settings.

This study highlights the prognostic significance of B7-H3 expression in localized renal cell carcinoma, demonstrating its strong correlation with disease progression and survival outcomes. The integration of molecular biomarkers such as B7-H3 into risk stratification tools, as exemplified by the RIPI, provides a promising avenue for enhancing personalized patient care. By incorporating tumor necrosis, pathological staging, and B7-H3 expression, the RIPI effectively identifies high-risk patients who may benefit from closer monitoring. In addition, considering the studies associating B7-H3 with immunotherapy resistance, further studies are necessary to use the RIPI both prognostically and predictively in the process of tailoring treatment in patients receiving adjuvant immunotherapy.

Despite the robust findings, it is important to acknowledge the limitations of this study. The retrospective design, single-center data, presence of selection bias, patients who did not receive adjuvant immunotherapy, relatively small sample size, lack of external validation, and lack of separate analyses within unclear subtypes are limitations that should be considered when interpreting the results.

In conclusion, B7-H3 stands out as both a prognostic biomarker and a potential therapeutic target in localized RCC. The RIPI represents a step forward in addressing the unmet need for precise risk stratification, paving the way for more tailored clinical management and improved patient outcomes in RCC.

## Figures and Tables

**Figure 1 medicina-61-00867-f001:**
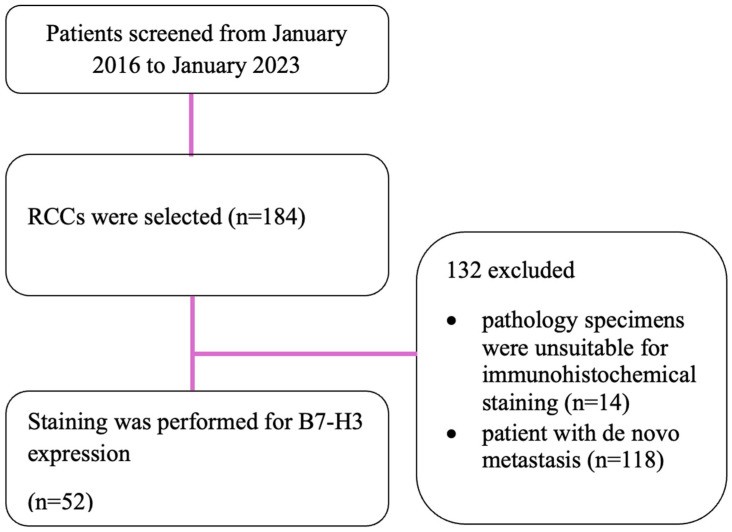
Patient flow chart.

**Figure 2 medicina-61-00867-f002:**
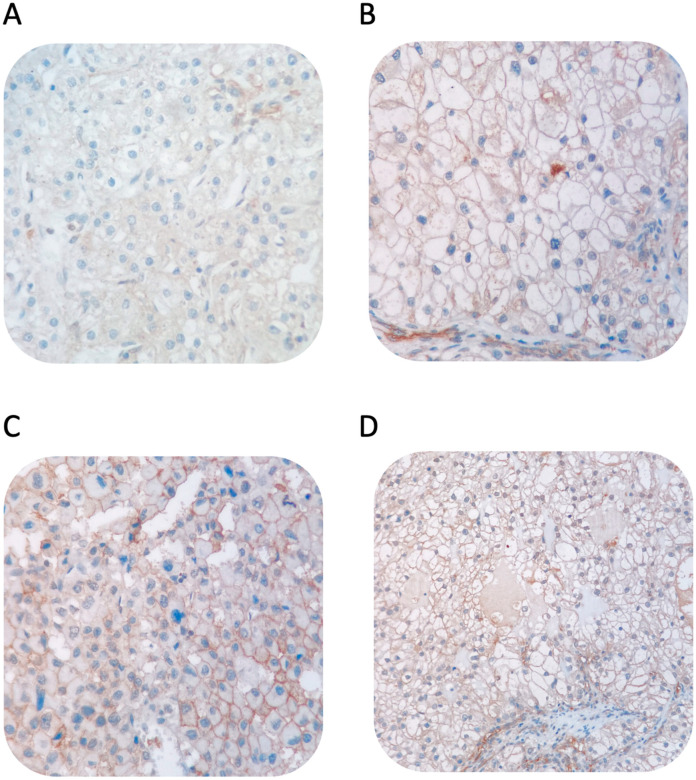
B7-H3 negative staining (**A**); B7-H3 +1 staining (**B**); B7-H3 +2 staining (**C**); B7-H3 +3 staining (**D**).

**Figure 3 medicina-61-00867-f003:**
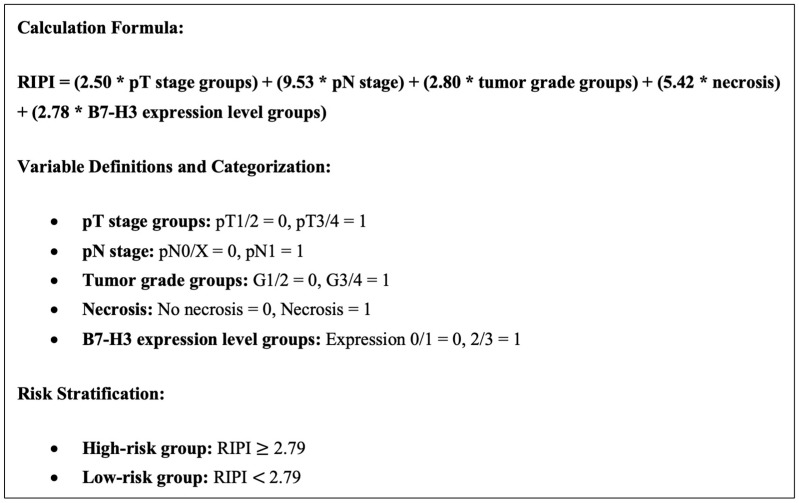
Renal Prognostic Immune Index (RIPI).

**Figure 4 medicina-61-00867-f004:**
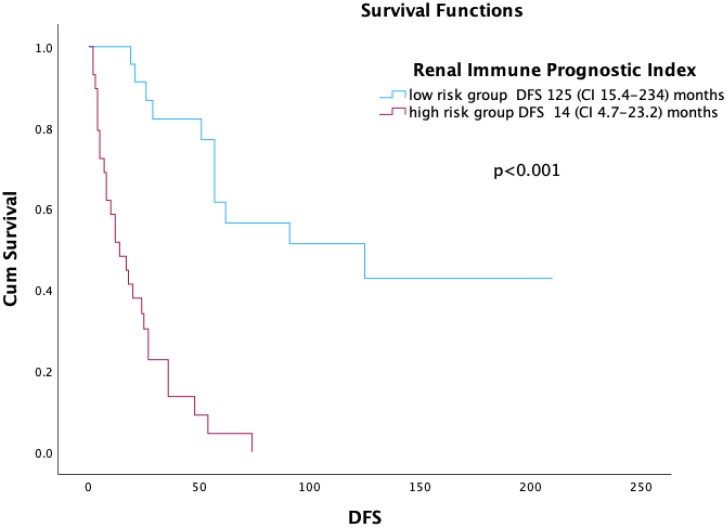
DFS according to RIPI.

**Table 1 medicina-61-00867-t001:** Patient characteristics.

		B7-H3 Expression Level Groups		*p* Value *
		0/1 (*n* = 36)	2/3 (*n* = 16)	
Age, years, mean		66 ± 10	62 ± 12	0.211
Sex, (%)	Female	10 (58.8)	7 (41.2)	0.257
	Male	26 (74.3)	9 (25.7)	
Karnofsky-PS (%)	≥70	31 (68.9)	14 (31.1)	0.633
	<70	5 (71.4)	2 (28.6)	
Tumor size, mm	Mean	67.9 ± 35.5	100.1 ± 48.1	0.075
Pathological T stage (%)	T1/T2	20 (80)	5 (20)	0.105
	T3/T4	16 (59.3)	11 (40.7)	
pN stage (%)	N0/Nx	35 (71.4)	14 (28.6)	0.165
	N1	1 (33.3)	2 (66.7)	
LVI	negative	30 (75.0)	10 (25.0)	0.100
	positive	6 (50.0)	6 (50.0)	
Pathological histology (%)	Clear-cell	24 (72.7)	9 (27.3)	0.393
	Non-clear-cell	12 (61.1)	7 (38.9)	
Tumor grade ^a^ (%)	G1/2	22 (78.5)	6 (21.5)	0.077
	G3/4	13 (56.5)	10 (43.5)	
Necrosis (%)	Negative	29 (80.6)	10 (26.3)	0.298
	Positive	7 (19.4)	6 (37.5)	
Sarcomatoid differentiation (%)	Negative	36 (73.5)	13 (26.5)	0.025 *
	Positive	0 (0)	3 (100)	

^a^: A chromophobe RCC was excluded. * *p* ≤ 0.05.

**Table 2 medicina-61-00867-t002:** Univariate and multivariate analyses for DFS.

	Univariate Analysis		Multivariate Analysis	
	HR (95%CI)	*p*-Value	HR (95%CI)	*p*-Value *
Age (<70 vs. >70)	1.18 (0.61–2.29)	0.64		
Sex (male vs. female)	1.27 (0.66–2.48)	0.47		
Karnofsky-PS (≥70 vs. <70)	2.06 (0.90–4.72)	0.087		
Pathological T stage (T1/T2 vs. T3/T4)	4.11 (2.02–8.35)	0.001	2.50 (1.08–5.80)	0.032
pN stage (N0/Nx vs. N1)	20.89 (4.19–104)	0.001	9.53 (1.77–51.1)	0.008
Pathological histology (clear-cell vs. non-clear-cell)	0.91 (0.46–1.82)	0.80		
Tumor grade (G1/2 vs. G3/4)	4.35 (2.14–8.83)	0.00 *	2.80 (1.27–6.17)	0.01
LVI (no vs. yes)	3.08 (1.24–6.38)	0.002		0.561
Necrosis (no vs. yes)	5.47 (2.47–12.1)	0.001	5.42 (2.12–13.8)	0.001
Sarcomatoid differentiation (no vs. yes)	2.47 (0.74–8.20)	0.137		
B7-H3 expression (low vs. high)	2.91 (1.49–5.66)	0.02	2.78 (1.35–5.70)	0.005

* *p* ≤ 0.05.

## Data Availability

The data that support the findings of this study are available on request from the corresponding author.

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
