# Peer review of "Prognostic Value of B7-H3 and a Novel Scoring System in Localized Renal Cell Carcinoma"

_medicina, 2025, doi:10.3390/medicina61050867_

Round 1

Reviewer 1 Report

Comments and Suggestions for Authors

This study by Faruk Recep Özalp et al. evaluated a new scoring system based on B7-H3 expression in combination with other clinicopathological factors for localized RCC. The manuscript is clearly written, with only a few minor revisions suggested:

  1. In Figure 2, panel D should be referenced in the figure legend.
  2. Although Table 1 specifies the criteria, it would help readers to also define low/high B7-H3 expression based on immunohistochemistry scores within the main text.
  3. Since antibody efficacy can vary by clone, it is recommended to include details (e.g., clone, catalog number, and manufacturer) for the B7-H3 antibody used.

Author Response

We appreciate the reviewer's positive comments and suggestions.

Comment 1: In Figure 2, panel D should be referenced in the figure legend.

Response 1: Panel D reference added to figure legend (Page 6).

Comment 2: Although Table 1 specifies the criteria, it would help readers to also define low/high B7-H3 expression based on immunohistochemistry scores within the main text.

Response 2: Low and high B7-H3 expression groups were added to the last paragraph under the subheading Immunohistochemical analysis in the Materials and methods section (Page 5).

Comment 3: Since antibody efficacy can vary by clone, it is recommended to include details (e.g., clone, catalog number, and manufacturer) for the B7-H3 antibody used.

Response 3: Details about the antibody have been added to the Materials and methods section (Page 5).(We have added this description: ANTI-CD276 EA, MA5-15693, Thermo Fisher Scientific, Waltham, MA USA)

Reviewer 2 Report

Comments and Suggestions for Authors

This study investigates the prognostic relevance of B7-H3 expression in localized renal cell carcinoma (RCC) and introduces a novel scoring system—the Renal Immune Prognostic Index (RIPI)—which combines molecular and pathological parameters to stratify recurrence risk. As an oncologist with a research interest in this area and a clear understanding of the unmet needs our patients face, I would like to commend the authors for their transparent and dedicated work. The incorporation of immune checkpoint-related biomarkers into clinical risk models represents a promising advancement in RCC prognostication. The manuscript provides meaningful insights; however, there are a few minor points that should be addressed prior to publication.

Comments

  1. The Introduction could be streamlined for clarity. Certain mechanistic or in-depth discussions currently placed in this section would be better suited for the Discussion, allowing the rationale of the study to stand out more clearly from the start. In the İntroduction, rather than broad mechanistic explorations, it may be more impactful to emphasize the heterogeneity of renal cell carcinoma subtypes, including the existence of tumors driven by distinct pathways such as immunogenic signaling or angiogenesis (vasculogenesis). This underscores the importance of precise patient selection in clinical management. The rationale for biomarker research can be better grounded by highlighting the need for better allocation of limited healthcare resources, with the aim of identifying patients most likely to benefit from intensive follow-up or adjuvant therapy. In this context, your study effectively positions B7-H3 as a promising, easily applicable immunohistochemical marker, and this point could be emphasized more directly as the central aim of the research.

  2.  In the Results section, it is stated that “B7-H3 was not detected” in certain cases. Could the authors please clarify whether this refers to a complete lack of staining (i.e., a true negative with a score of 0), or whether B7-H3 could not be evaluated due to technical issues such as poor tissue preservation, lack of internal controls, or suboptimal staining? If the latter, it would be important to explain how such cases were handled in the final analysis—were they excluded, scored as negative, or imputed in some way? Clarifying this point is crucial, especially for readers aiming to understand the reproducibility and clinical applicability of the proposed scoring system.

  3. The inclusion of patients with different RCC histologies (clear cell, papillary, chromophobe, etc.) is reasonable, but subgroup outcomes (especially for non-clear cell RCC) are not deeply analyzed. Given the biological differences between subtypes, consider discussing how RIPI performs across these groups or acknowledging this as a limitation.

4. The study has already correctly identified the potential of B7-H3 as a therapeutic target, but its role as a resistance marker for checkpoint inhibitors is only briefly mentioned. This connection could be further elaborated, especially as B7-H3-directed therapies (e.g., enoblituzumab) are entering clinical evaluation.

5. The retrospective nature and exclusion of patients receiving adjuvant immunotherapy limit the immediate translation of RIPI into clinical trial stratification. Please clarify how RIPI could be applied or validated in the setting of adjuvant pembrolizumab, particularly for the high-risk group identified in this study.

Minor Comments

  1. The manuscript is generally clear but would benefit from minor English editing. For example, the sentence “Similarly, grade 3/4 tumours were associated with significantly DFS (12 months) compared to grade 1/2 tumours (57 months, p < 0.001)” is grammatically incorrect and unclear. It should be revised as:
    1. “Similarly, grade 3/4 tumours were associated with significantly shorter DFS (12 months) compared to grade 1/2 tumours (57 months, p < 0.001).”

  2. The PET and immunohistochemical figures are visually helpful. 
  3. While the study is well contextualized, please expand the section on limitations, especially regarding selection bias and lack of external validation.

This manuscript addresses an important and evolving area in RCC prognostication by combining immune biomarker data with classic pathology to create a refined risk model. The development and application of RIPI are promising, but further clarification is needed regarding its validation, generalizability, and clinical integration. 

Author Response

Thank you for all the comments and suggestions of the reviewer.

Comment 1: The Introduction could be streamlined for clarity. Certain mechanistic or in-depth discussions currently placed in this section would be better suited for the Discussion, allowing the rationale of the study to stand out more clearly from the start. In the Ä°ntroduction, rather than broad mechanistic explorations, it may be more impactful to emphasize the heterogeneity of renal cell carcinoma subtypes, including the existence of tumors driven by distinct pathways such as immunogenic signaling or angiogenesis (vasculogenesis). This underscores the importance of precise patient selection in clinical management. The rationale for biomarker research can be better grounded by highlighting the need for better allocation of limited healthcare resources, with the aim of identifying patients most likely to benefit from intensive follow-up or adjuvant therapy. In this context, your study effectively positions B7-H3 as a promising, easily applicable immunohistochemical marker, and this point could be emphasized more directly as the central aim of the research.

Response 1: Thanks for your suggestions. We have improved paragraph 3 in the introduction by addressing the utility of biomarkers for the effective use of limited resources (Page 3, Lines 71-77). We have moved the paragraph including references 18 and 19 in the introduction to paragraph 2 in the discussion section (Page 10-11, Lines 233-238).

Comment 2: In the Results section, it is stated that “B7-H3 was not detected” in certain cases. Could the authors please clarify whether this refers to a complete lack of staining (i.e., a true negative with a score of 0), or whether B7-H3 could not be evaluated due to technical issues such as poor tissue preservation, lack of internal controls, or suboptimal staining? If the latter, it would be important to explain how such cases were handled in the final analysis—were they excluded, scored as negative, or imputed in some way? Clarifying this point is crucial, especially for readers aiming to understand the reproducibility and clinical applicability of the proposed scoring system.

Response 2: The statement refers to true zero scores. We changed the sentence by removing the word ‘detected’ which may cause misunderstanding. Patients who could not be evaluated due to technical problems (n=14) are reported in the materials and methods section.(After revision: In 31% (n=16) of the cases, B7-H3 expression was 0, while the expression levels in the remaining samples were as follows: 38.5% (n=20) of the participants scored +1, 25% (n=13) +2 and 5.7% (n=3) +3)(Page 7, Lines 169-171).

Comment  3: The inclusion of patients with different RCC histologies (clear cell, papillary, chromophobe, etc.) is reasonable, but subgroup outcomes (especially for non-clear cell RCC) are not deeply analyzed. Given the biological differences between subtypes, consider discussing how RIPI performs across these groups or acknowledging this as a limitation.

Response 3: Due to the small sample size, we were able to analyse as clear vs. nonclear RCC. The lack of nonclear RCC subgroup analysis was added to the limitations.(After revision: Retrospective design, single centre data, presence of selection bias, patients who did not receive adjuvant immunotherapy, relatively small sample size, lack of external validation, lack of separate analyses within unclear subtypes are limitations that should be considered when interpreting the results.) (Page 12, Lines 299-303).

Comment 4: The study has already correctly identified the potential of B7-H3 as a therapeutic target, but its role as a resistance marker for checkpoint inhibitors is only briefly mentioned. This connection could be further elaborated, especially as B7-H3-directed therapies (e.g., enoblituzumab) are entering clinical evaluation.

Response 4: We did not discuss specific agents because the patients had localised disease and current targeted therapies were used at the earliest 2nd line.

Comment 5: The retrospective nature and exclusion of patients receiving adjuvant immunotherapy limit the immediate translation of RIPI into clinical trial stratification. Please clarify how RIPI could be applied or validated in the setting of adjuvant pembrolizumab, particularly for the high-risk group identified in this study.

Response 5:The relationship between B7-H3 expression and DFS should be evaluated in a group patients receiving adjuvant immunotherapy prior to RIPI validation. It is important to note that patients in the high-risk group who receive adjuvant pembrolizumab after RIPI validation may not experience the best prognosis. In such cases, it may be worthwhile to consider prolongation of pembrolizumab treatment/dual immune checkpoint therapy or the use of B7-H3 targeted therapies.  If, however, it appears that pembrolizumab is not providing benefit in the high-risk RIPI group, then it may be advisable to consider follow-up without treatment. (This comment was not added to the article in detail. The following sentence was added to the Discussion: In addition, given the studies associating B7-H3 with immunotherapy resistance, further studies are needed to use RIPI both prognostically and predictively in the process of tailoring treatment in patients receiving adjuvant immunotherapy)(Page 12, Lines 295-298).   Minor Comments

Comment 1: The manuscript is generally clear but would benefit from minor English editing. For example, the sentence “Similarly, grade 3/4 tumours were associated with significantly DFS (12 months) compared to grade 1/2 tumours (57 months, p < 0.001)” is grammatically incorrect and unclear. It should be revised as:

    1. “Similarly, grade 3/4 tumours were associated with significantly shorter DFS (12 months) compared to grade 1/2 tumours (57 months, p < 0.001).”

Response 1: We have thoroughly proofread the manuscript and corrected the grammatical issues identified. The specific sentence cited has been revised as suggested. Additionally, minor language improvements were made throughout the manuscript.

Comment 2: The PET and immunohistochemical figures are visually helpful. 

Response 2: Thank you for your kind comment.

Comment 3: While the study is well contextualized, please expand the section on limitations, especially regarding selection bias and lack of external validation.

Response 3: We have expanded the Limitations section to more fully discuss the retrospective design, potential selection bias, small sample size and lack of external validation as we outlined in Response 3 above, and these points are now explicitly acknowledged in the revised Discussion section.